# Screening for undiagnosed non-alcoholic fatty liver disease (NAFLD) and non-alcoholic steatohepatitis (NASH): A population-based risk factor assessment using vibration controlled transient elastography (VCTE)

Wayne Eskridge[1]*, John M. Vierling[2], Wayne Gosbee[3], Gabriella A. Wan[1], May-Linh Hyunh[1], Henry E. Chang[1]

**1** Fatty Liver Foundation, Boise, Idaho, United States of America, **2** Section of Gastroenterology and Hepatology and Division of Abdominal Transplantation, Departments of Medicine and Surgery, Baylor College of Medicine, Houston, Texas, United States of America, **3** Link2Labs, Houston, Texas, United States of America

☯ These authors contributed equally to this work.
* wayne@fattyliverfoundation.org

## Abstract

The screening for undiagnosed non-alcoholic fatty liver disease and non-alcoholic steatohepatitis (SUNN) study was a population-based screening study that aimed to provide proof of concept to encourage community-level screening and detection for this non-communicable disease. Current screening guidelines do not recommend the routine screening of nonalcoholic fatty liver disease (NAFLD) for asymptomatic populations, so providers are not encouraged to actively seek disease, even in high-risk patients. This study sought to determine whether a self-selecting cohort of asymptomatic individuals would have scores based on vibration controlled transient elastography (VCTE) and controlled attenuation parameter (CAP) significantly correlated to risk factors to suggest that routine screening for high-risk patients should be recommended. The study recruited 1,070 self-selected participants in Houston and Galveston County, Texas, 940 of which were included in final analysis. A pre-screening survey was used to determine eligibility. VCTE-based scores analyzed steatosis and fibrosis levels. Fifty-seven percent of the study population demonstrated steatosis without fibrosis, suggesting NAFLD, while 16% demonstrated both steatosis and fibrosis, suggesting NASH. Statistically significant risk factors included factors related to metabolic syndrome, race, and age, while statistically significant protective factors included consumption of certain foods and exercise. The findings of this study suggest that high-risk individuals should be screened for NAFLD even in the absence of symptoms and that community-based screenings are an effective tool, particularly in the absence of proactive guidelines for providers.

**Data Availability Statement:** All relevant data are within the paper and its Supporting Information files.

**Funding:** We received study grants from Intercept Pharmaceuticals (https://www.interceptpharma.com/) and the Eskridge Family Trust. In-kind contributions were made by Health Business Solutions. The opinions, results and conclusions reported in this paper are those of the authors. The funders had no role in study design, data collection and analysis, decision to publish, or preparation of the manuscript.

**Competing interests:** I have read the journal's policy and the authors of this manuscript have the following competing interests: WE, GAW, MLH, and HEC are employees of FLF, which reports program support and educational grants from Allergan, Amazon, Bristol-Myers Squibb, Celgene, Clinical Care Options, Continuum Clinical, Echosens, Eskridge Family Trust, Fibronostics, First Line Creative, Gilead Sciences, Global Engage, Google, Health Business Solutions, Intercept Pharmaceuticals, Madrigal Pharmaceuticals, Meetrix, Merck & Co., Inc. NetNoggin, Prosciento, Pfizer, Terns Pharmaceuticals and various private and philanthropic individual donors. JMV reports relevant research and grant support from Allergan, Alnylam, 89Bio, Bristol-Myers Squibb, Celgene, CymaBay, Exalenz, Galectin, Galmed, Genfit, Gilead, Hanmi, Immuron, Intercept, Madrigal, Merck, Mochida, Molecular Stethoscope, Novartis, NovoNordisk, Pliant, Sagimet, Tobira Therapeutics, and Zydus; advisor or consultant fees from Allergan, Blade Pharmaceuticals, Bristol-Myers Squibb, Conatus, CymaBay, Exalenz, Fractyl, Intercept, Novartis; and is on the Data Safety and Management Boards for NIH NIDDK Drug-Induced Liver Injury Network (DILIN) and Fractyl. WG reports grant support from AbbVie and Gilead Sciences. This does not alter our adherence to PLOS ONE policies on sharing data and materials.

## Introduction

Non-alcoholic fatty liver disease (NAFLD) is the most common form of chronic liver disease worldwide with a global prevalence of 25% [1]. One-third of American adults are thought to have NAFLD, which occurs when excess fat builds up in liver cells, also known as steatosis [2]. NAFLD can progress to non-alcoholic steatohepatitis (NASH) characterized by hepatic inflammation, ballooning degeneration of hepatocytes, and progressive fibrosis [3]. Progressive NASH culminates in cirrhosis with future risks for complications of portal hypertension, liver failure, and hepatocellular carcinoma (HCC) [4]. In the US, adult liver transplants are predominantly performed for alcohol-related liver disease and NASH [5]. In a recent study comparing the epidemiological trends of various liver diseases in the US over the past three decades, NAFLD was the only one with a consistently increasing prevalence, increasing by 20% from 1988–1994 to 31.9% from 2013–2016 and mirroring the rise in type 2 diabetes mellitus (T2DM), obesity, insulin resistance, and hypertension [6].

Currently, the American Association for the Study of Liver Disease (AASLD) does not recommend the routine screening of NAFLD among asymptomatic populations, even if they are high-risk [7]. In contrast, clinical practice guidelines from the European Association for the Study of the Liver (EASL) and from the Asian Pacific Association for the Study of the Liver (APASL) recommend or suggest considering screening for patients who are obese or who have type 2 diabetes mellitus (T2DM) [8]. Without screening for NAFLD, it is both unlikely and difficult to detect patients with NASH and significant fibrosis, as most are asymptomatic and many do not have elevated serum aminotransferase levels. If screening is performed, the first-line assessment for steatosis is typically ultrasound due to its low cost and wide availability, even though it is not the most sensitive of tests [8]. The current gold standard for diagnosis is liver biopsy, a procedure that is costly, invasive, and can lead to complications such as pain, minor or major bleeding, and even death [9]. Furthermore, liver biopsy is prone to sampling error, inter-observer variability, and poor acceptance by patients [10]. Many noninvasive tests are emerging as alternatives to ultrasound and biopsy, including blood based tests such as FIB-4 and LIVERFASt™; and imaging tests such as VCTE (FibroScan® or Velacur™), magnetic resonance elastography (MRE), and LiverMultiScan®. Because MRE and LiverMultiScan® require magnetic resonance imaging (MRI) they remain expensive compared to VCTE options [11]. Combination of VCTE with CAP score allows for semi-quantification of hepatic steatosis and liver stiffness, the surrogate for fibrosis [11]. VCTE and CAP for screening of NAFLD/NASH has demonstrated significant usefulness when studied in pediatric patients and could use further validation in the adult population [12].

In terms of treatment, the absence of approved pharmaceutical treatments for NAFLD or NASH limits recommendations to lifestyle modifications to promote weight loss, such as exercise and a nutritious diet, and treatments for features of metabolic syndrome [7]. Metabolic management is appropriate for those with suspected NASH and F0-F1 fibrosis, while those with NASH and F2 or greater fibrosis should be referred to a hepatologist and encouraged to consider enrolling in clinical trials for new therapies [7]. It is important to recognize that NAFLD and NASH are part of a spectrum of metabolic comorbid diseases that significantly increase all-cause mortality [13]. Thus, NAFLD is considered the hepatic manifestation of metabolic syndrome (MetS) [9] that includes obesity, T2DM, high triglyceride levels, high LDL cholesterol, and hypertension [14]. NAFLD contributes to the development and progression of cardiovascular disease (CVD), T2DM, chronic kidney disease (CKD), obstructive sleep apnea (OSA), and extra-hepatic malignancies (e.g., colorectal cancer) [13,15]. Because many of these diseases share co-dependent risk factors, successful management of NAFLD can improve comorbid diseases and vice versa [13]. As global noncommunicable disease rates continue to

rise, screening high-risk populations and ensuring diagnosed individuals are making the appropriate lifestyle interventions to slow down or prevent disease progression are increasingly important.

A key question of this research is who among this large pool of patients should be prioritized for NAFLD screening? Cost effective analysis indicates that screening all obese persons is not effective. Though prevalence of F2 or greater fibrosis is significant enough for screening to be cost effective, the lack of effective treatment besides lifestyle and diet change makes it difficult for screening to have a positive effect on patient outcomes. Experts in the field have expressed opinions indicating that as medications for the treatment of NAFLD and NASH become available, screening will indeed become cost effective [16].

The complicated nature of NAFLD has presented some challenges in the field. Even with liver biopsy as the accepted gold standard, no clear consensus exists for diagnosis of NAFLD by specialists (e.g. gastroenterologists, hepatologists, endocrinologists, cardiologists, oncologists) and primary care providers. Concerns persist about how to decide if a patient needs a biopsy, or which non-invasive diagnostic methods would be most effective at detecting disease [17]. The four categories of histological criteria used to grade and stage NAFLD are fibrosis, steatosis, inflammation, and cellular ballooning, the last of which has shown weak interobserver agreement [17], complicating assuredness in the accuracy of diagnosis and disease management. Furthermore, as NAFLD progresses to NASH, disease profiles can become increasingly heterogenous, and things such as lipoprotein patterns may be more or less important in different patients [17]. In addition to lifestyle, individual traits such as ethnicity and genetics can also contribute to disease manifestation, making NAFLD a disease affected by both environmental and genetic factors and signifying that treatment may need to be approached from a multifactorial perspective. Lastly, the notion that NAFLD is fatty liver disease in the absence of alcohol consumption is less straightforward than it seems, particularly because the definitions of mild and moderate consumption are inconsistent [18]. Studies have shown both a protective [19–21] and detrimental [22–24] effect of mild and moderate alcohol consumption on NAFLD, leaving room for interpretation and further investigation.

## Study overview

This study aimed to demonstrate that an undiagnosed asymptomatic population of NAFLD and NASH patients existed and would opt to be proactive about their health when presented the opportunity, and to elucidate risk and protective factors among this population. Though many studies around the world have investigated the prevalence and natural history of NAFLD using population-based methods, broad scale population-based screening for NAFLD has only been performed in children [25–29]. By deploying noninvasive screening to a self-selecting population-based cohort, this study sought to demonstrate that current guidelines for diagnosis are not adequately detecting the true disease burden. It aimed to include around 1000 patients to ensure an adequately powered statistical analysis within the constraints of available funding and was successful in achieving its aims.

## Materials and methods

In December 2018, the SUNN Study set out to challenge the AASLD's current screening guidance recommendation by initiating a community screening program in South Houston and Galveston County, Texas to demonstrate disease prevalence in asymptomatic individuals. Even though ultrasound is a typical first line screening tool, this study did not employ ultrasound for two reasons: it is less reliable when hepatic steatosis is less than 20% and it is not a portable device that can be easily brought to community sites [30]. Instead, VCTE and CAP

were used to identify participants with non-progressive or slowly progressive cirrhosis as well as participants with steatosis and fibrosis most likely attributable to NASH. Individuals whose VCTE and CAP scores indicated steatosis and fibrosis required further testing to exclude causes of liver disease before a making a formal diagnosis of NASH and subsequent referral to appropriate care [7]. A registered nurse was hired to perform screening tests and temporary staff members provided programmatic support. Day-to-day activities that centered around outreach included recruitment, venue identification, screening, and distribution of relevant information to study participants. The study was approved by WIRB (Work Order Number 1-1117038-1, IRB Tracking Number 20182311). Written informed consent was obtained from all participants.

## Recruitment

Recruitment and advertising for screening events occurred using Google Ads, through social media, and by placing informational materials on local bulletin boards in physician offices, churches, and other areas of community gatherings. Participants were self-selecting and no personally identifiable information was collected. A pre-screening survey was used to collect demographic and health history information and to disqualify those who were not eligible to participate. Data collected included information about metabolic conditions, other health conditions, diet and exercise habits, alcohol intake, as well as the presence of other liver diseases, and can be found in S1 File. Reasons for ineligibility were too large of a distance from the FibroScan® XL probe to capsule, known history of liver disease, and significant alcohol consumption per week (defined by AASLD as more than 14 drinks for females and more than 21 drinks for males) [7]. Fig 1 shows the study participant selection process; a total of 1,070 participants were recruited and 940 were included in analysis.

## Screening

Screening events were held at various community-based facilities throughout the local area including health fairs, the Mexican Consulate General in Houston, Federally Qualified Health Centers (FQHCs), the Galveston County Health Department, and local service groups such as the Domestic Workers Association and small business employee meetings. Participants were pre-screened in one area, and if eligible to participate in the study proceeded to a private screening room.

To perform the screening for NAFLD/NASH, researchers used a portable FibroScan® machine (Echosens™ North America, Waltham, MA, USA). FibroScan® is a non-invasive imaging device that measures liver fat content (steatosis) and liver stiffness (fibrosis) that can be found in clinical settings throughout the US. It is an in-person procedure similar to an ultrasound that takes approximately 15 minutes to complete. The test results include a transient elastography (TE) score and CAP score. TE measures fibrosis in kilopascals (kPa), with scores ranging from 1.5 to 75 kPa, where lower values indicate higher liver elasticity [31]. The ranges of TE scores used for this analysis were categorized into fibrosis grade according to Table 1, with cutoffs adapted from the Memorial Sloan Kettering Cancer Center guidelines [32]. The CAP score measures steatosis in decibels per meter (dB/m), with scores ranging from 100 to 400 dB/m. The ranges of CAP scores used for this analysis were categorized into steatosis grade according to Table 2, with cutoffs adopted from the same Memorial Sloan Kettering Cancer Center guidelines as TE Scores [32].

As diagnostic tools, TE and CAP are not robust enough to accurately define the stages being described, as there can be much overlap in interpretation. Conducting a liver biopsy is the current gold standard to diagnose NAFLD. However, given its cost and burden, this study

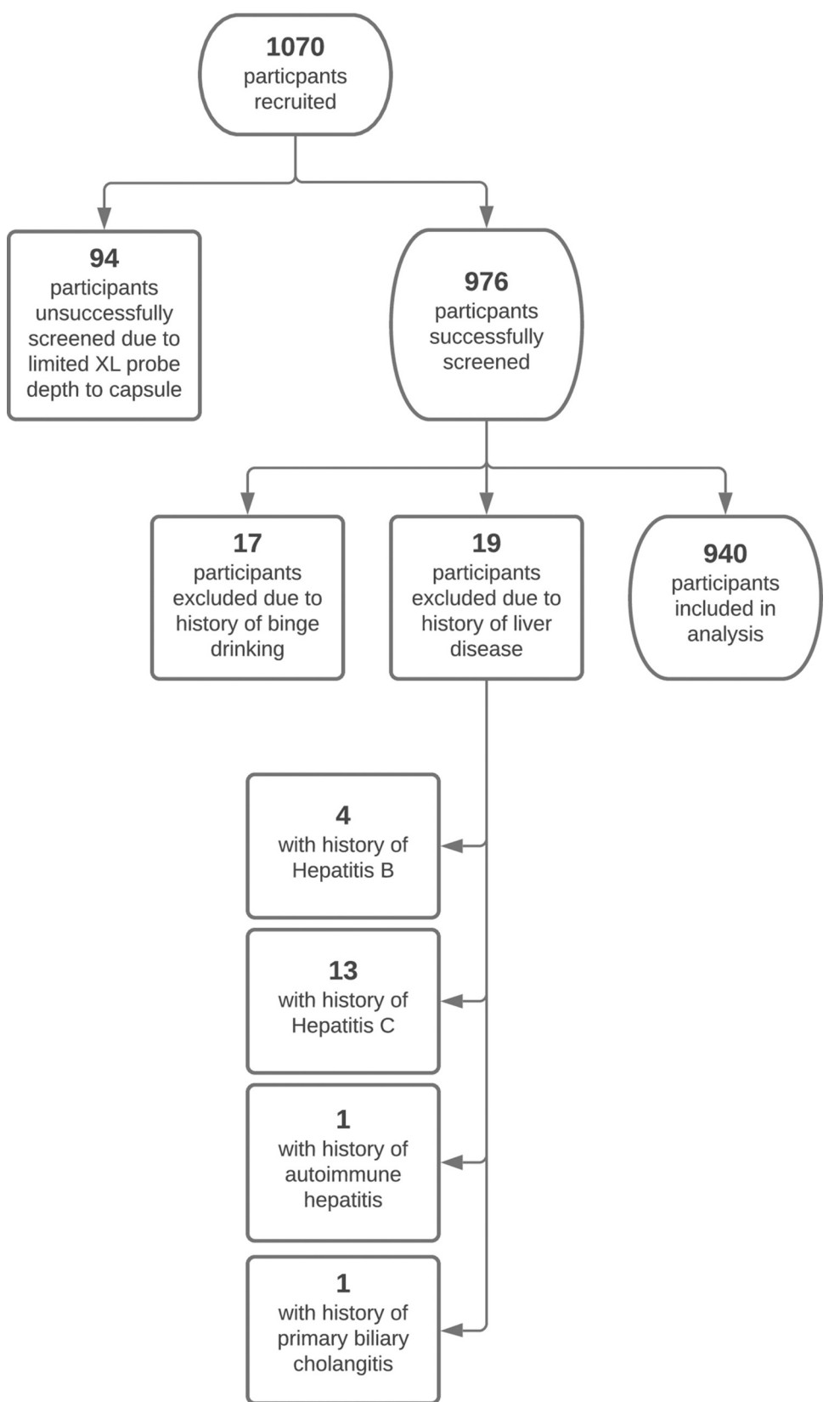

**Fig 1. Participant selection.** Study participant selection process.

**Table 1. Transient elastography (TE) scores.**

|  | F0 No Fibrosis | F1 Mild fibrosis | F2 Moderate fibrosis | F3 Severe fibrosis | F4 Cirrhosis |
|---|---|---|---|---|---|
| TE (kPa) | 0 to ≤7.0 | >7 to ≤7.5 | >7.5 to ≤10.0 | >10 to ≤14 | >14.0 |

used the noninvasively acquired staging with FibroScan® as the method of stratification. Participants were categorized into four groups based on fibrosis and steatosis categorization of the liver: no fat and no stiffness, no fat with stiffness, fat without stiffness, and fat with stiffness. Though a well-recognized and validated tool, these measurements of TE and CAP alone do not represent a diagnosis of disease. Therefore, no official diagnoses were made as a result of this study. Even so, the TE and CAP scores obtained through FibroScan® allowed the researchers to identify patients for whom follow up with a medical provider was warranted. After being screened, participants received their scores, a packet of information about NAFLD and NASH, and contact information for additional resources. Participants with elevated scores were strongly advised to seek follow-up care with their primary care provider or a hepatologist.

## Statistical analyses

To identify those at-risk for advancing asymptomatic liver disease, FibroScan® scores were analyzed for indicated fibrosis and steatosis. To complete the analysis, steatosis grade S3 was used as a proxy diagnosis for NAFLD and fibrosis grades of F3 and F4 were used as a proxy diagnosis for NASH. Survey variables were height, weight, sex, age, race, BMI category, diabetes, high blood pressure, high cholesterol, high triglycerides, swollen joints, heart disease, heart attack, irregular heartbeat, stroke, arthritis, weak or broken bones, low thyroid, low vitamin D, low testosterone, in menopause, Crohn's or ulcerative colitis, cancer, protein consumption, starch consumption, vegetable consumption, fat consumption, nut consumption, sugar consumption, fast food meals, restaurant meals, cardiovascular exercise, strength training exercise, alcohol consumption, and other liver disease. All continuous variables were made categorical and Pearson's chi-square tests were conducted to identify statistical significance as related to TE and CAP scores. After identifying variables with statistically significant relationships to TE and CAP scores, odds ratio analyses were conducted using logistic regression to sort the variables as either risk or protective factors. Complete raw data can be found in S2 File, and missing values are noted in S3 File. Calculations were made using Stata SE (Version 16.1) and figures were made using Lucidchart. A p-value <0.05 was considered statistically significant.

## Results

### Demographics

A demographic breakdown of the study population can be seen in Table 3. Of the 940 participants included in analysis, 67% (n = 626) identified as female and 33% (n = 314) identified as male. The mean age of participants was 47.58; the youngest participant was 18 and the oldest participant was 89. Approximately 60% (n = 581) of study participants identified as Hispanic, 15% (n = 147) identified as African American, 14% (n = 132) identified as Caucasian, 4%

**Table 2. Controlled attenuation parameter (CAP) scores.**

|  | S0 No steatosis | S1 Mild steatosis | S2 Moderate steatosis | S3 Severe steatosis |
|---|---|---|---|---|
| CAP (dB/m) | 0 to <238 | 238 to ≤260 | 260 to ≤290 | >290 |

**Table 3. Demographics of study population.**

| Demographics | Frequency | Percent |
|---|---|---|
| Age | | |
| 18–19 years | 11 | 1.17 |
| 20–29 years | 87 | 9.26 |
| 30–39 years | 159 | 16.91 |
| 40–49 years | 258 | 27.45 |
| 50–59 years | 237 | 25.21 |
| 60–69 years | 133 | 14.15 |
| 70–79 years | 38 | 4.04 |
| 80+ years | 10 | 1.06 |
| Missing | 7 | 0.74 |
| Sex | | |
| Female | 626 | 66.60 |
| Male | 313 | 33.30 |
| Missing | 1 | 0.00 |
| Race | | |
| White | 133 | 14.15 |
| Black | 147 | 15.64 |
| Asian | 43 | 4.57 |
| Hispanic | 581 | 61.81 |
| Other | 17 | 1.81 |
| Mixed | 5 | 0.53 |
| Missing | 14 | 1.49 |
| BMI | | |
| Underweight | 7 | 0.74 |
| Normal | 184 | 19.57 |
| Overweight | 327 | 34.79 |
| Obese | 421 | 44.79 |
| Missing | 1 | 0.00 |
| Prevalence of Comorbidities | | |
| Diabetes | 247 | 26.28 |
| High blood pressure | 262 | 27.87 |
| High cholesterol | 264 | 28.09 |
| High triglycerides | 118 | 12.55 |
| Swollen joints | 62 | 6.60 |
| Heart disease | 20 | 2.13 |
| Heart attack | 15 | 1.60 |
| Irregular heartbeat | 36 | 3.83 |
| Stroke | 19 | 2.02 |
| Arthritis | 114 | 12.13 |
| Weak/broken bones | 38 | 4.04 |
| Low thyroid | 89 | 9.47 |
| Low vitamin D | 135 | 14.36 |
| Low testosterone | 38 | 4.04 |
| In menopause | 115 | 12.23 |
| Crohn's/colitis | 22 | 2.34 |
| Skin cancer (melanoma) | 5 | 0.53 |
| Skin cancer (basal/squamous cell) | 16 | 1.70 |

(*Continued*)

**Table 3.** (Continued)

| Demographics | Frequency | Percent |
|---|---|---|
| Other cancers | 34 | 3.62 |
| Alcohol consumption (drinks/week) | | |
| None | 599 | 63.72 |
| 1–2 | 169 | 17.98 |
| 3–4 | 68 | 7.23 |
| 5–6 | 32 | 3.40 |
| 7–8 | 18 | 1.91 |
| 9–12 | 37 | 3.94 |
| 13–14 | 11 | 1.17 |
| 15–20 | 6 | 0.64 |

(n = 43) identified as Asian, and 1.5% (n = 15) of participants did not disclose their race. The majority of participants were overweight or obese, and the most commonly reported comorbidities were diabetes, high blood pressure, and high cholesterol. The majority of participants, 64% (n = 599), reported having zero drinks per week.

## Primary analysis: Prevalence of fibrosis and steatosis

The primary analysis of this study was to determine the population prevalence of fibrosis and steatosis. As seen in Table 4, the majority of the study population, 57% (n = 540), had fibrosis and steatosis scores that indicated liver fat without liver stiffness; 24% (n = 230) indicated no fat and no stiffness; 16% (n = 150) indicated fat with stiffness; and 2% (n = 20) of participants indicated no fat with stiffness. Table 5 provides a detailed breakdown of steatosis and fibrosis score distribution.

From these FibroScan® results, approximately 40% (n = 374) of the study population indicated a steatosis grade of S3, meeting this study's criteria for NAFLD. Approximately 5% (n = 49) of the study population had a fibrosis score of F3 or F4, meeting this study's criteria for NASH.

## Secondary analysis: Survey variables

The secondary analysis of this study was to determine correlation between demographic and lifestyle variables with proxy diagnoses of NAFLD and NASH. Survey variables were investigated to determine statistically significant relationships to the populations with a proxy NAFLD diagnosis, defined as elevated CAP scores (S3); and the populations with a proxy NASH diagnosis, defined as elevated TE scores (F3/F4). From the analysis, 12 unique variables were identified as having a statistically significant relationship with either proxy NAFLD (10 statistically significant variables or proxy NASH (5 statistically significant variables). 3 of the variables showed statistical significance for both proxy NAFLD and proxy NASH. The statistically significant variables are outlined in Table 6.

**Table 4. Breakdown of study population based on fibrosis and steatosis categorization.**

| Characteristics | Fibrosis score (TE Value) | Steatosis score (CAP Value) | n | % of total |
|---|---|---|---|---|
| No fat and no stiffness | F0 ($\leq$ 7.0 kPa) | S0 (<238 dB/m) | 230 | 24.47 |
| No fat with stiffness | F1-F4 (>7.0 kPa) | S0 (<238 dB/m) | 20 | 2.13 |
| Fat without stiffness | F0 ($\leq$ 7.0 kPa) | S1-S3 (>238 dB/m) | 540 | 57.45 |
| Fat with stiffness | F1-F4 (>7.0 kPa) | S1-S3 (>238 dB/m) | 150 | 15.96 |

**Table 5. Steatosis and fibrosis score distribution.**

| n %    | S0              | S1              | S2              | S3              | Total           |
|--------|-----------------|-----------------|-----------------|-----------------|-----------------|
| F0     | 230 24.47%      | 117 12.45%      | 155 16.49%      | 268 28.51%      | 770 81.91%      |
| F1     | 8 0.85%         | 5 0.52%         | 13 1.36%        | 17 1.78%        | 43 4.57%        |
| F2     | 9 0.94%         | 3 0.31%         | 15 1.57%        | 51 5.33%        | 78 8.30%        |
| F3     | 2 0.21%         | 1 0.10%         | 2 0.21%         | 25 2.61%        | 30 3.19%        |
| F4     | 1 0.10%         | 1 0.10%         | 4 0.42%         | 13 1.36%        | 19 2.02%        |
| Total  | 250 26.59%      | 127 13.51%      | 189 20.11%      | 374 39.79%      | **940 100.00%** |

Among the population with a proxy NAFLD diagnosis in this study (n = 374), 67% were clinically obese (BMI greater than 30), 38% were diabetic, 34% had hypertension, 33% had high cholesterol, and 16% had high triglyceride levels. Among the population with a proxy NASH diagnosis in this study (n = 49), 90% were clinically obese, 45% were diabetic, 45% had hypertension, 6% had a history of stroke, and 35% had a history of arthritis.

**Table 6. Variables with statistically significant relationships to proxy NAFLD and NASH diagnoses.**

| Significant variables with proxy NAFLD | Significant variables with proxy NASH |
|---|---|
| Body mass index* | Body mass index* |
| Overweight | Overweight |
| Obesity | Obesity |
| Diabetes* | Diabetes* |
| Hypertension* | Hypertension* |
| High cholesterol | Stroke |
| High triglycerides | Arthritis |
| Age | |
| 40-49 years | |
| 50–59 years | |
| Race | |
| Asian | |
| Hispanic | |
| Vegetable consumption | |
| 3–4 days per week | |
| 5–6 days per week | |
| 7 days per week | |
| Nut consumption | |
| 3–4 days per week | |
| 5–6 days per week | |
| Strength training exercise | |
| 1–2 days per week | |
| 3–4 days per week | |
| 5–6 days per week | |

*overlapping variables.

**Table 7. Odds ratio analysis of statistically significant variables for TE and CAP scores.**

| Variable | Variable subgroup | Odds Ratio | 95% CI | p-value |
|---|---|---|---|---|
| *Variables significantly related to proxy NAFLD (S3)* | | | | |
| Body mass index* | Overweight | 2.76 | 1.70–4.48 | 0.000 |
| | Obesity | 9.21 | 5.79–14.65 | 0.000 |
| Diabetes* | Yes | 2.69 | 2.00–3.62 | 0.000 |
| Hypertension* | Yes | 1.64 | 1.23–2.19 | 0.001 |
| High cholesterol | Yes | 1.51 | 1.13–2.01 | 0.005 |
| High triglycerides | Yes | 1.74 | 1.18–2.56 | 0.005 |
| Age | 40–49 years | 1.89 | 1.12–3.18 | 0.017 |
| | 50–59 years | 1.84 | 1.08–3.10 | 0.024 |
| Race | Asian | 2.10 | 1.03–4.25 | 0.040 |
| | Hispanic | 1.91 | 1.27–2.87 | 0.002 |
| Vegetable consumption | 3–4 days per week | 0.42 | 0.18–0.98 | 0.044 |
| | 5–6 days per week | 0.24 | 0.10–0.58 | 0.001 |
| | 7 days per week | 0.32 | 0.13–0.74 | 0.008 |
| Nut consumption | 3–4 days per week | 0.57 | 0.37–0.88 | 0.011 |
| | 5–6 days per week | 0.53 | 0.29–0.95 | 0.032 |
| Strength training exercise | 1–2 days per week | 0.64 | 0.46–0.90 | 0.01 |
| | 3–4 days per week | 0.50 | 0.30–0.84 | 0.009 |
| | 5–6 days per week | 0.34 | 0.14–0.79 | 0.012 |
| *Variables significantly related to proxy NASH (F3/F4)* | | | | |
| Body mass index* | Overweight | 2.47 | 1.12–5.47 | 0.026 |
| | Obesity | 9.61 | 4.59–20.11 | 0.000 |
| Diabetes* | Yes | 2.41 | 1.35–4.32 | 0.003 |
| Hypertension* | Yes | 2.21 | 1.23–3.96 | 0.008 |
| Stroke | Yes | 3.57 | 1.00–12.68 | 0.049 |
| Arthritis | Yes | 4.35 | 2.33–8.12 | 0.000 |

**Odds ratio analysis.** An odds ratio (OR) analysis using logistic regression was conducted for the variables found to be statistically significantly correlated to proxy NAFLD and NASH diagnoses. An OR greater than 1 indicated a risk factor, an OR <1 indicated a protective factor, and an OR = 1 indicated no significant relationship. 9 variables were identified as risk factors and 2 variables were identified as protective factors. The results of the OR analysis are summarized in Table 7.

**Risk factors.** Risk factors for a proxy NAFLD diagnosis were race, age, BMI, diabetes, hypertension, high cholesterol, and high triglycerides. Compared to their White counterparts, participants identifying as Hispanic (OR 1.91, 95%CI 1.27–2.87, p = 0.002) and Asian (OR 2.10, 95%CI 1.03–4.125, p = 0.040) were roughly doubly at risk for a proxy NAFLD diagnosis. Compared to participants aged 20 to 29 years old, those between 40 and 49 years old had 1.89 (95%CI 1.12–3.18, p = 0.017) higher odds of having a proxy NAFLD diagnosis, while those between 50 and 59 were 1.84 (95%CI 1.08–3.10, p = 0.024) times more likely to have a proxy NAFLD diagnosis. Participants who were overweight were 2.76 (95%CI 1.70–4.48, p<0.001) times more at risk to have a proxy NAFLD diagnosis than those who were normal weight, and those who were obese were 9.21 (95%CI 5.79–14.65, p<0.001) times more at risk for a proxy NAFLD diagnosis. Participants with diabetes were 2.69 (95%CI 2.00–3.62, p<0.001) times more likely to have a proxy NAFLD diagnosis than those without diabetes. Participants with hypertension had a 1.64 (95%CI 1.29–2.19, p = 0.001) times greater risk of a proxy NAFLD diagnosis than those without hypertension. Participants with high cholesterol were 1.51 (95%

CI 1.13–2.01, p = 0.005) times more likely to have a proxy NAFLD diagnoses and participants with high triglycerides were at 1.74 (95%CI 1.18–2.56, p = 0.005) times greater risk of having a proxy NAFLD diagnosis.

Risk factors for individuals with a proxy NASH diagnosis were BMI, diabetes, hypertension, history of stroke, and arthritis. Compared to individuals of normal weight, participants who were clinically overweight were 2.47 (95%CI 1.12–5.47, p = 0.026) times more at risk of having a fibrosis score of F1 or greater, and participants who were obese were 9.61 (95%CI 4.59–20.11, p<0.001) times more at risk. Odds ratio analysis for BMI was analyzed for risk of F1 and higher to avoid issues of collinearity. Participants with diabetes were 2.41 (95%CI 1.35–4.32, p = 0.003) times more likely to have a proxy NASH diagnosis than those without diabetes. Participants with hypertension were 2.21 (95%CI 1.23–3.96, p = 0.008) times more likely to have proxy NASH diagnosis than those without hypertension, while a history of stroke increased the odds of a proxy NASH diagnosis by a factor of 3.57 (95%CI 1.00–12.68, p = 0.049). Compared to participants without arthritis, those with arthritis were 4.35 (95% CI2.33–8.12, p<0.001) times more at risk for a proxy NASH diagnosis.

**Protectiv*e* factors.** The three protective factors identified against having a proxy NAFLD diagnosis were frequency of non-starchy vegetable consumption, frequency of nut consumption, and frequency of strength training exercise. Participants who ate non-starchy vegetables for more than 3 days per week had lower odds of having a proxy NAFLD diagnosis than those who did not eat vegetables at all. Participants who reported eating non-starchy vegetables 3–4 days per week were 0.42 (95%CI 0.18–0.98, p = 0.044) times as likely to have a proxy NAFLD diagnosis, demonstrating a reduced risk of 58%. Those who reported eating non-starchy vegetables 5–6 days per week were 0.24 (95%CI 0.10–0.58, p = 0.001) times as likely to have a proxy NAFLD diagnosis, reducing their risk by 76%. Those who reported eating non-starchy vegetables 7 days per week were 0.32 (95%CI 0.13–0.74, p = 0.008) times as likely to have a proxy NAFLD diagnosis, a 68% reduction of risk. Similarly, semi-regular nut consumption also demonstrated reduced risk for a proxy NAFLD diagnosis. Compared to those who reported never eating nuts, those who ate nuts 3–4 days per week were 0.57 (0.37–0.88, p = 0.011) times as likely to have a proxy NAFLD diagnosis, a 43% reduced risk, and those who ate nuts 5–6 days per week were 0.53 (95%CI 0.29–0.95, p = 0.032) times as likely to have a proxy NAFLD diagnosis, a 47% reduced risk. For strength training exercise, frequency demonstrated an inverse relationship with risk. Compared to those who reported no engagement in strength training exercise, participants who reported engaging in strength training exercise 1–2 days per week were 0.64 (95%CI 0.46–0.90, p = 0.01) times as likely to have a proxy NAFLD diagnosis, reducing their risk by 36%. Those who reported engaging in strength training exercise 3–4 days per week were 0.5 (95%CI 0.30–0.89, p = 0.009) times as likely to have a proxy NAFLD diagnosis, a 50% reduction in risk. Most notably, those who reported engaging in strength training exercise 5–6 days per week were only 0.34 (95%CI 0.14–0.79, p = 0.012) times as likely to have a proxy NAFLD diagnosis, reducing their risk by 64%.

No protective factors were identified against having a proxy definition of NASH in this study population.

## Discussion

In this study, about one quarter (24%) of the study population had no indicated fat or liver stiffness. A majority of the study population (57%) indicated liver fat without liver stiffness, while participants with both liver fat and liver stiffness made up approximately 16% of the study population. Approximately 40% (n = 374) of the study population met this study's criteria for NAFLD and approximately 5% (n = 49) of the study population met this study's criteria

for NASH. As estimates for prevalence of NAFLD in the US range from 10% to 30% [2], estimated prevalence in the study population was roughly double that of the general population. It is important to recognize that participants in the study enrolled on a self-selection basis. Participants were not directed by their physicians to get screened, but rather chose to opt-in to learn about their liver health and potential risk factors when presented the opportunity. As a result, it is possible that the study population was enriched with a higher prevalence for NAFLD and NASH than the general population because the individuals recruited were concerned enough about their health and potential risk factors to volunteer for screening. A key characteristic of the study population was that participants wanted to understand their potential disease status when presented with the opportunity and may have been more motivated than the general population to be proactive about their health.

BMI indicating overweight and obese, diabetes, and hypertension were all risk factors for proxy diagnoses of NAFLD and NASH. These lifestyle and comorbid conditions are the only three variables that showed statistically significant relationships with both proxy NAFLD and proxy NASH definitions, indicating that they increased risk for both incidence of NAFLD and progression to NASH. Though the mechanism of action for the transition of NAFLD to NASH is not well understood, it is helpful to understand that high BMI, diabetes, and hypertension contribute to risk at both stages. Therefore, targeting screening to these populations may have the biggest potential to reduce the burden of NAFLD and NASH in the short-term.

Race also proved to be an important risk factor, as individuals in this study who identified as Hispanic and Asian had higher risks of having a proxy NAFLD diagnosis. A genotype study conducted in 2008 demonstrated that the *I148M* allele variant of the *PNPLA3* gene showed that individuals who are homozygous carriers of the allele have greater levels of fat in their liver than noncarriers [33]. By 2019, several genome-wide association studies had firmly established *PNPLA3 I148M* as a genetic modifier of steatosis in the liver and a risk factor for steatohepatitis, fibrosis, and HCC [34,35]. These studies also established that individuals of Hispanic and Asian ancestry are more likely to carry the genetic variant, a finding that is suggested by the results of this study.

Although age was a risk factor for a proxy NAFLD diagnosis, it is unclear whether older individuals are more susceptible to NAFLD or whether disease progression had advanced sufficiently over time to be detectable via screening measures. Nonetheless, age is a risk factor and thus should be considered when identifying potential screening populations. It is particularly important to recognize that those aged 40–49 were at the highest risk of progressive disease, as this population is younger than might be expected. The lower incidence in the 50–59 group seen in this study of asymptomatic patients with no known liver disease may reflect the emergence of symptoms and the disqualification of that population for this study.

An individual's diet contributed to his or her risk of a proxy NAFLD diagnosis. An inverse relationship was seen between a diet of non-starchy vegetables and steatosis grade. As vegetable consumption increased, the odds ratio decreased, demonstrating a protective effect. Participants who reported eating non-starchy vegetables 7 days per week had a reduced risk of having a proxy NAFLD diagnosis by 68% when compared to individuals who did not eat vegetables any day of the week. Similarly, those who ate nuts regularly had 43–47% reduced risk of a proxy NAFLD diagnosis. These findings contribute to current studies that show a relationship between healthier diets and lower incidence of NAFLD, suggesting that individuals who are at-risk for NAFLD begin incorporating non-starchy vegetables and nuts into their weekly routines [36].

Leading an active lifestyle also had a positive effect on risk of a proxy NAFLD diagnosis. An inverse relationship was seen between strength training and steatosis grade: compared to those who did no strength training at all, individuals who participated in strength training 1–2 days per week had a reduced risk of having a proxy NAFLD diagnosis by 36%, while those who exercised 3–4 days per week saw a reduced risk of 50%, and those who exercised 5–6 days per

week had a reduced risk of 64%. The more frequently an individual reported strength training, the less likely they were to have proxy NAFLD diagnosis. This information contributes to current studies that demonstrate a relationship between exercise and a lower incidence of NAFLD, suggesting that individuals who are at-risk for NAFLD should begin integrating physical activity and exercise into their weekly routines [35].

This study demonstrated a significant burden of NAFLD and NASH among self-selected volunteers in a community setting. In light of the AASLD guidelines that do not recommend screening in asymptomatic populations, this study demonstrated that a large disease burden remains undetected in the population. Community-based screening efforts create an opportunity for concerned and high-risk individuals to become informed about their potential disease status and can spark a teachable moment to encourage care seeking behavior from providers who would be otherwise unlikely to recommend screening, diagnosis, and disease management. Screening efforts targeted to those with high BMIs, diabetes, or high blood pressure could have the greatest impact in reducing disease burden in the near future, as these risk factors are relevant to both the proxy NAFLD and proxy NASH diagnoses in this study. This study also provides evidence to support current practices for managing NAFLD progression through lifestyle modifications such as diet, exercise, and weight loss. Specifically, the findings of this study suggest that individuals who incorporated strength training, non-starchy vegetables, and nuts into one's weekly routine had a lower risk of a proxy NAFLD diagnosis.

## Strengths and limitations

This study had several strengths. The community-based model allowed for a broad sample of the general population to be captured. Approaching people at community locations, largely outside of the formal healthcare setting, allowed patients with unmet health needs to be reached in a way that was not intimidating, obligating, or overwhelming. By creating a teachable moment, community-based screenings may have motivated individuals to be more aware of their liver health, encouraged healthy behaviors for those deemed to have or be at-risk for NAFLD/NASH, and increased the chances of regular doctor visits for previously unknown conditions such as NAFLD/NASH.

This study also had a few limitations. Firstly, because the study population consisted of individuals who chose to participate, self-selection bias is a feature. The population screened was presumed to have been more aware of their risk or more likely to be proactive health seekers. A key question in this study's design, as researchers were going into the community and offering screening to anyone, was whether a significant pool of at-risk but asymptomatic patients would voluntarily seek testing to justify screening. Another limitation of this study is that there are currently no well-established cutoff values for CAP to indicate NAFLD. Because of this lack of standardization, several studies have provided original cutoff values for CAP and steatosis grade [37–41]. Discrepancies between cutoff values may relate to difference in study designs and populations, including disease etiologies, the prevalence of obesity and subcutaneous adiposity, and the severity of steatosis, which all may influence CAP performance. The final limitation of this study was the inability to track uptake of services after the screening was performed. Future community-based screening studies for NAFLD/NASH should aim to capture data regarding linkage to care after screening.

## Conclusion

This study demonstrates that non-invasive imaging screening with TE and CAP of asymptomatic persons with self-identified risk factors for NAFLD or NASH is both feasible and acceptable to participants. Providing immediate information about steatosis alone or steatosis and

fibrosis indicative of NASH to participants increased their awareness of NAFLD and clarified their personal need for further medical evaluation. Voluntary non-invasive imaging screening with TE and CAP of persons with risk factors for NAFLD and NASH can identify those with a risk of progressive disease and need for subsequent medical care. Furthermore, community-based screening of a self-selected at-risk population demonstrated a disease prevalence roughly double the prevalence estimated for the general population, suggesting that targeted screening could significantly raise awareness of NAFLD/NASH and help encourage early interventions among those most at-risk for progression.

## Supporting information

**S1 File. Participant survey.** Demographics and past medical history survey questions. (PDF)

**S2 File. Raw data.** Stata file containing all raw data analyzed. (DTA)

**S3 File. Summary statistics.** Excel file containing all analyzed summary statistics. (XLSX)

## Acknowledgments

The authors would like to express particular gratitude to all individuals who volunteered their time and participated in this study. The authors would also like to thank Laura Mosely for serving as the study's RN, as well as the Galveston County Health District, Consulate General of Mexico in Houston, Galveston County Domestic Workers Alliance, Glad Tidings of Texas City, multiple Federally Qualified Health Centers in Houston and Texas City, and multiple health fairs for their invaluable support in the promotion of this study and facilitation in the recruitment of study participants.

## Author Contributions

**Data curation:** Wayne Eskridge, John M. Vierling.

**Formal analysis:** John M. Vierling, Gabriella A. Wan, May-Linh Hyunh.

**Funding acquisition:** Wayne Eskridge.

**Investigation:** Wayne Eskridge, John M. Vierling, Wayne Gosbee.

**Methodology:** Wayne Eskridge, John M. Vierling.

**Project administration:** Wayne Eskridge, Wayne Gosbee.

**Resources:** Wayne Eskridge, Wayne Gosbee.

**Software:** Wayne Eskridge, Gabriella A. Wan, May-Linh Hyunh.

**Supervision:** Wayne Eskridge, Wayne Gosbee.

**Validation:** John M. Vierling, Gabriella A. Wan.

**Visualization:** Gabriella A. Wan, May-Linh Hyunh.

**Writing – original draft:** May-Linh Hyunh.

**Writing – review & editing:** Wayne Eskridge, John M. Vierling, Gabriella A. Wan, Henry E. Chang.

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
