## [Decision Letter · Decision Letter 0]

16 Aug 2021

PONE-D-21-19005

Screening for undiagnosed non-alcoholic fatty liver disease (NAFLD) and non-alcoholic steatohepatitis (NASH): a population-based risk factor assessment using vibration controlled transient elastography (VCTE)

PLOS ONE

Dear Dr. Eskridge,

Thank you for submitting your manuscript to PLOS ONE. After careful consideration, we feel that it has merit but does not fully meet PLOS ONE’s publication criteria as it currently stands. Therefore, we invite you to submit a revised version of the manuscript that addresses the points raised during the review process.

We look forward to receiving your revised manuscript.

Kind regards,

Daisuke Tokuhara

Academic Editor

PLOS ONE

Journal Requirements:

2. Please provide additional details regarding participant consent. In the ethics statement in the Methods and online submission information, please ensure that you have specified whether consent was informed

"This study was funded by grants from the Eskridge Family Trust and Intercept Pharmaceuticals, Inc., with in-kind contributions from Health Business Solutions"

"WE received a study grants from Intercept Pharmaceuticals (https://www.interceptpharma.com/) and the Eskridge Family Trust. The opinions, results and conclusions reported in this paper are those of the authors. The funders had no role in study design, data collection and analysis, decision to publish, or preparation of the manuscript"

"WE received a study grants from Intercept Pharmaceuticals (https://www.interceptpharma.com/) and the Eskridge Family Trust. The opinions, results and conclusions reported in this paper are those of the authors. The funders had no role in study design, data collection and analysis, decision to publish, or preparation of the manuscript" 

Additional Editor Comments (if provided):

I deeply appreciate the authors' sincere efforts for the development of screening tool for NAFLD. Reviewer provided the constructive comments for the manuscript. I hope those comments will help authors to prepare for the revised manuscript.

Reviewers' comments:

Reviewer's Responses to Questions

**Comments to the Author**

1. Is the manuscript technically sound, and do the data support the conclusions?

Reviewer #1: Partly

Reviewer #2: Yes

2. Has the statistical analysis been performed appropriately and rigorously? 

Reviewer #1: No

Reviewer #2: Yes

3. Have the authors made all data underlying the findings in their manuscript fully available?

Reviewer #1: Yes

Reviewer #2: Yes

4. Is the manuscript presented in an intelligible fashion and written in standard English?

Reviewer #1: Yes

Reviewer #2: Yes

5. Review Comments to the Author

Reviewer #1: The current manuscript describes a self-selected population based study to identify patients with liver steatosis and/or liver fibrosis with use of transient elastography (TE) combined with controlled attenuation parameter (CAP). The aim of the study was to identify participants with steatosis and cirrhosis or fibrosis attributable to NASH, with the ultimate goal to demonstrate that community based screening can raise awareness, increased uptake of screening and affect behavior changes in (asymptomatic) NAFLD/NASH patients. The study included 940 patients of which baseline characteristics, co-morbidity and lifestyle variables were collected, further all patients underwent TE with CAP measurements to classify whether patients have NAFLD (steatosis S3), NASH (Fibrosis F3-4) or no abnormalities. In total 57% of the participants had steatosis without fibrosis (NAFLD) while 16% of patients had both fibrosis and steatosis (NASH). Identified risk factors in this study correspond with risk factors identified in literature. This paper contains usefull data for clinical practice, however there are some issues that need to be addressed by the authors

Major comments

I have difficulties with the various categories and definitions used in this paper for different analyses, which makes interpretation confusing. For example Table 4 presents four categories of steatosis and stiffness where S1 steatosis and F1 fibrosis are used. Later NAFLD is defined as having steatosis S3 and NASH is defined as having fibrosis F3-4 (which is defined in as steatosis and fibrosis in the abstract) and it is unclear if the NAFLD patients have no or mild fibrosis and whether the NASH patients can also have no steatosis (F0)? Further the logistic regression analysis seems to be performed on S3 steatosis (NAFLD) and F3-4 fibrosis (NASH) however in the results is F1 described, so this is unclear as well. I would advise the authors to define NAFLD en NASH based on literature and based on both steatosis and fibrosis stages and then perform all analyses on these stages.

Furthermore the manuscript is moderately written, not conform the usual outline of a paper. For example some methods are described in the results, the statistical analyses section contains a paragraph about collecting co-morbidity and life style and no strengths and limitations are described in the discussion. Also the introduction is too long and not concise, try to make a logical story here.

Minor comments

Abstract

- The authors describe 3 different aims, please describe more clearly

- 38: 1070 participants is misleading as 940 participants were included

Introduction

- Please try to be more concise in the introduction, the introduction now is far too long in my opinion

- What is the knowledge gap, has this type of population based screening been done before and what are the results?

- The aim of the study or research questions should be clearly presented in the last paragraph of the introduction

Study overview

- The aim is community based screening however this study is performed in self-selected participants which has a high risk of bias, this should be mentioned in the manuscript

Methods

- Screening: If I understand correct the authors used transient elastography with controlled attenuation parameter (CAP), please describe this clearly

- The gold standard for detecting steatosis based on EASL guidelines is ultrasound, can the authors explain why ultrasound is not used and describe this in the manuscript?

- In reading the statistical analyses paragraph there are some questions that come to my mind such as: Did the authors correct for multiple testing? Why did the authors perform chi square and later logistic regression? Was it univariate of multivariate regression analysis?

- Why did the authors include around 1000 patients, was this calculated or was there a pre-defined time schedule?

Results

- The baseline characteristics table (table 3) does not contain all information, please add all the other collected variables such as alcohol consumption, co-morbidity, BMI to describe the population in more detail etcetera.

- 237: Patients were categorized, this is methods

- Table 4: in the methods only F3-4 are described as relevant and only S3. However in these categories also F1 is described as fibrosis while TE performes bad in this category. See the advice in major comments.

- What is the explanation for patients with stiffness without fat

- 268: clinically obese, how is this defined?

- Table 6: This table is not necessary as the odds ratios are described in table 7

- Table 7: please use NASH and NAFLD in the grey rows, or describe the degree of fibrosis and steatosis.

- 287: Here F1 is mentioned while I understood that the analyses were performed on participants with F3-4 fibrosis? How are the analyses done?

Discussion

- 371: Participants 40-49 year were at highest risk for what? NAFLD or NASH? Please describe more clear.

- 382 – 387 repetition of results, please be more concise and interpret the findings.

- Strenghts and limitations are missing

- 393: This study shows an association between diet and steatosis and fibrosis, however it does not show that it lowers the risk. Also, it can be questioned how precise this data is as both physical activity and diet can change from week to week.

Typo

- 39: ‘were’ should be removed

- 213: Crohn instead of Chron, ulcerative colitis instead of colitis

- Table 3: please change ‘blank’ into ‘missing’

Reviewer #2: I totally agreed with the proposal that high-risk individuals should be screened for NAFLD by means of VCTE even in the absence of symptoms and that community-based screenings by using VCTE are an effective tool.

Comment 1: Current study clearly demonstrated the usefulness of VCTE for the screening of NAFLD in adult population. A previous study (Cho Y, et al. Plos One. 2015. 10. e0137239) have also demonstrated the significance of VCTE in the assessment of NAFLD in pediatric patietns. I recommend to cite this previous paper that will further strengthen the significance of authors' proposal.

Comment 2: I have another comment should be addressed as a study limitaion in the revised manuscript. The most significant and difficult issue to be discussed is the cut off and/or reference value of CAP to determine NAFLD. Authors used the cut off value of CAP for determining the presence of NAFLD based on the previous study (Zhang X et al. Clin Mol Hepatol. 2020). However there are currently no established cut-off value of CAP of NAFLD. Numerous studies provided or used original cut off or reference value of CAP for NAFLD and healthy individuals in adult, children and adolescents. Therefore I strongly recommend to cite at least the following references from different countries in the revised manuscript. Then authors should add the study limitation in terms of the use of the cut off or reference value of CAP. For example, discrepancies of the cut off and/or reference value may relate to differences in the study desin and populations including disease aetiologies, the prevalence of obesity and extent of subcutaneous adiposity, and the severity of steatosis, which may influence CAP performance.

(Reference 1) Sasso M, et al. Ultrasound Med Biol. 2010;36(11):1825-35.

(Reference 2) de Lédinghen V, et al. Liver Int. 2012;32(6):911-8.

(Reference 3) Tokuhara D, et al. Plos ONE. 2016;11:e0166683

(Reference 4) Isoura Y, et al. Obes Res Clin Pract. 2020;14(5):473-478

(Reference 5) Chon YE, et al. Liver Int. 2014;34(1):102-9.

6. PLOS authors have the option to publish the peer review history of their article (what does this mean?). If published, this will include your full peer review and any attached files.

Reviewer #1: No

Reviewer #2: No

---

## [Author Response · Author response to Decision Letter 0]

29 Sep 2021

Journal Requirements:

• Lines 373 and 376: italicized gene name

• Lines 592-595: Renamed Supporting information Filess

2. Please provide additional details regarding participant consent. In the ethics statement in the Methods and online submission information, please ensure that you have specified whether consent was informed

• Line 150: changed “written consent” to “written informed consent”

"This study was funded by grants from the Eskridge Family Trust and Intercept Pharmaceuticals, Inc., with in-kind contributions from Health Business Solutions"

"WE received a study grants from Intercept Pharmaceuticals (https://www.interceptpharma.com/) and the Eskridge Family Trust. The opinions, results and conclusions reported in this paper are those of the authors. The funders had no role in study design, data collection and analysis, decision to publish, or preparation of the manuscript"

• Lines 467-468: removed “This study was funded by grants from the Eskridge Family Trust and Intercept Pharmaceuticals, Inc., with in-kind contributions from Health Business Solutions.”

• Funding statement should read: “WE received study grants from Intercept Pharmaceuticals (https://www.interceptpharma.com/) and the Eskridge Family Trust. In-kind contributions were made by Health Business Solutions. The opinions, results and conclusions reported in this paper are those of the authors. The funders had no role in study design, data collection and analysis, decision to publish, or preparation of the manuscript."

"WE received a study grants from Intercept Pharmaceuticals (https://www.interceptpharma.com/) and the Eskridge Family Trust. The opinions, results and conclusions reported in this paper are those of the authors. The funders had no role in study design, data collection and analysis, decision to publish, or preparation of the manuscript" 

• Intercept Pharmaceuticals

• Eskridge Family Trust

• Health Business Solutions

• No authors received a salary from any of our funders.

• The authors received specific funding from the above sources for this work.

Additional Editor Comments (if provided):

I deeply appreciate the authors' sincere efforts for the development of screening tool for NAFLD. Reviewer provided the constructive comments for the manuscript. I hope those comments will help authors to prepare for the revised manuscript.

Reviewer #1: The current manuscript describes a self-selected population based study to identify patients with liver steatosis and/or liver fibrosis with use of transient elastography (TE) combined with controlled attenuation parameter (CAP). The aim of the study was to identify participants with steatosis and cirrhosis or fibrosis attributable to NASH, with the ultimate goal to demonstrate that community based screening can raise awareness, increased uptake of screening and affect behavior changes in (asymptomatic) NAFLD/NASH patients. The study included 940 patients of which baseline characteristics, co-morbidity and lifestyle variables were collected, further all patients underwent TE with CAP measurements to classify whether patients have NAFLD (steatosis S3), NASH (Fibrosis F3-4) or no abnormalities. In total 57% of the participants had steatosis without fibrosis (NAFLD) while 16% of patients had both fibrosis and steatosis (NASH). Identified risk factors in this study correspond with risk factors identified in literature. This paper contains usefull data for clinical practice, however there are some issues that need to be addressed by the authors

Major comments

I have difficulties with the various categories and definitions used in this paper for different analyses, which makes interpretation confusing. For example Table 4 presents four categories of steatosis and stiffness where S1 steatosis and F1 fibrosis are used. Later NAFLD is defined as having steatosis S3 and NASH is defined as having fibrosis F3-4 (which is defined in as steatosis and fibrosis in the abstract) and it is unclear if the NAFLD patients have no or mild fibrosis and whether the NASH patients can also have no steatosis (F0)? Further the logistic regression analysis seems to be performed on S3 steatosis (NAFLD) and F3-4 fibrosis (NASH) however in the results is F1 described, so this is unclear as well. I would advise the authors to define NAFLD en NASH based on literature and based on both steatosis and fibrosis stages and then perform all analyses on these stages.

• I understand the confusion of this reviewer, but I disagree that analysis needs to be redone. We consulted with an independent biostatistician for his opinion, and he agreed that the analysis was effective and appropriate. I have edited the text and tables to make the analysis clearer. Firstly, I more explicitly defined the primary and secondary analyses that were performed. From the primary analysis, I removed the breakdown of the “fat with stiffness” population in Table 4 and the corresponding text (Lines 245-247), as this seemed to confuse the reviewer as a different definition of NAFLD/NASH. To further elucidate the secondary analysis, I changed the descriptions of elevated TE and elevated CAP scores to be referred to as proxy NAFLD and proxy NASH, respectively. The tables were relabeled and rearranged to first analyze NAFLD then NASH. Language was adapted throughout the discussion to mirror the changes and make the analysis and findings easier to understand.

Furthermore the manuscript is moderately written, not conform the usual outline of a paper. For example some methods are described in the results, the statistical analyses section contains a paragraph about collecting co-morbidity and life style and no strengths and limitations are described in the discussion. Also the introduction is too long and not concise, try to make a logical story here.

Minor comments

Abstract

- The authors describe 3 different aims, please describe more clearly

• Removed original Lines 34-37, clarifying that the aim of the study was to demonstrate enough disease prevalence and correlation of risks in an asymptomatic population to warrant more proactive screening recommendations.

- 38: 1070 participants is misleading as 940 participants were included

• Line 36 (previously Line 38): edited to explain 940 were included in final analysis

Introduction

- Please try to be more concise in the introduction, the introduction now is far too long in my opinion

• Removed redundant and extraneous sections.

- What is the knowledge gap, has this type of population based screening been done before and what are the results?

• Lines 128-130: added in details about previous population-based research

- The aim of the study or research questions should be clearly presented in the last paragraph of the introduction

• Lines 123-132: The “Study Overview” section, which already presents the aim, is the last paragraph of the introduction

Study overview

- The aim is community based screening however this study is performed in self-selected participants which has a high risk of bias, this should be mentioned in the manuscript

• Line 432-437: self-selection bias is a feature of this research; one of the questions the research aimed to answer was if a significant pool of at-risk asymptomatic patents would voluntarily seek testing to justify the need for broad screening. Volunteering participants, and therefore self-selection bias, are part of the model and cannot be avoided.

Methods

- Screening: If I understand correct the authors used transient elastography with controlled attenuation parameter (CAP), please describe this clearly

• Lines 179-187: TE and CAP descriptions already included

- The gold standard for detecting steatosis based on EASL guidelines is ultrasound, can the authors explain why ultrasound is not used and describe this in the manuscript?

• Lines 138-141: added rationale for not using ultrasound

- In reading the statistical analyses paragraph there are some questions that come to my mind such as: Did the authors correct for multiple testing? Why did the authors perform chi square and later logistic regression? Was it univariate of multivariate regression analysis?

- Why did the authors include around 1000 patients, was this calculated or was there a pre-defined time schedule?

• Lines 130-132: incorporated rationale for study size

Results

- The baseline characteristics table (table 3) does not contain all information, please add all the other collected variables such as alcohol consumption, co-morbidity, BMI to describe the population in more detail etcetera.

• Line 238 (Table 3): added rows for avg BMI, co-morbidities, alcohol consumption

- 237: Patients were categorized, this is methods

• Line 241 (previously 237): “participants we categorized…fat with stiffness,” moved to methods section lines 196-198 

- Table 4: in the methods only F3-4 are described as relevant and only S3. However in these categories also F1 is described as fibrosis while TE performes bad in this category. See the advice in major comments.

• See response to major comments

- What is the explanation for patients with stiffness without fat

• Lines 209-211: patients with F3-4 stiffness were considered NASH, even without fat. Patients with no fat and F1-2 stiffness had no proxy diagnosis.

- 268: clinically obese, how is this defined?

• Line 273 (previously 268): defined clinically obese as BMI >30

- Table 6: This table is not necessary as the odds ratios are described in table 7

• Line 268-269 (Table 6): This table is remaining because the overlap between statistically significant variables for TE and CAP scores makes it confusing to list in the text. The findings are presented more clearly in table format than they would be listed in the text.

- Table 7: please use NASH and NAFLD in the grey rows, or describe the degree of fibrosis and steatosis.

• Line 285 (Table 7): Degree of fibrosis and steatosis described in gray rows

- 287: Here F1 is mentioned while I understood that the analyses were performed on participants with F3-4 fibrosis? How are the analyses done?

• Line 307 (previously 287): F1 is mentioned here because OR analysis for BMI (overweight, obese, normal weight) was analyzed for risk of fibrosis score of F1 or greater to avoid issues of collinearity, as stated in lines 308-309.

Discussion

- 371: Participants 40-49 year were at highest risk for what? NAFLD or NASH? Please describe more clear.

• Line 384-385 (previously 371): edited to be clearer that 40 to 49-year-olds were at highest risk for progressive disease

- 382 – 387 repetition of results, please be more concise and interpret the findings.

• Lines 396-401 (previously Lines 382-387): there is no repetition of results, just demonstration of the inverse relationship; a sentence was added lines 382-384 as interpretation of the findings

- Strenghts and limitations are missing

• Lines 421-443: added section for strengths and limitations

- 393: This study shows an association between diet and steatosis and fibrosis, however it does not show that it lowers the risk. Also, it can be questioned how precise this data is as both physical activity and diet can change from week to week.

• Line 402-405 (previously Line 393): reworded to reflect association of findings instead of causation

Typo

- 39: ‘were’ should be removed

• Line 36 (previously Line 39): removed errant ‘were’ (previously in line 39)

- 213: Crohn instead of Chron, ulcerative colitis instead of colitis

• Line 214 (previously Line 213): changed to Crohn’s and ulcerative colitis

- Table 3: please change ‘blank’ into ‘missing’

• Line 238 (Table 3): Changed ‘blank’ to ‘missing’

Reviewer #2: I totally agreed with the proposal that high-risk individuals should be screened for NAFLD by means of VCTE even in the absence of symptoms and that community-based screenings by using VCTE are an effective tool.

Comment 1: Current study clearly demonstrated the usefulness of VCTE for the screening of NAFLD in adult population. A previous study (Cho Y, et al. Plos One. 2015. 10. e0137239) have also demonstrated the significance of VCTE in the assessment of NAFLD in pediatric patietns. I recommend to cite this previous paper that will further strengthen the significance of authors' proposal.

• Line 78-80: added text referencing VCTE for NAFLD assessment in pediatric populations

Comment 2: I have another comment should be addressed as a study limitaion in the revised manuscript. The most significant and difficult issue to be discussed is the cut off and/or reference value of CAP to determine NAFLD. Authors used the cut off value of CAP for determining the presence of NAFLD based on the previous study (Zhang X et al. Clin Mol Hepatol. 2020). However there are currently no established cut-off value of CAP of NAFLD. Numerous studies provided or used original cut off or reference value of CAP for NAFLD and healthy individuals in adult, children and adolescents. Therefore I strongly recommend to cite at least the following references from different countries in the revised manuscript. Then authors should add the study limitation in terms of the use of the cut off or reference value of CAP. For example, discrepancies of the cut off and/or reference value may relate to differences in the study desin and populations including disease aetiologies, the prevalence of obesity and extent of subcutaneous adiposity, and the severity of steatosis, which may influence CAP performance.

(Reference 1) Sasso M, et al. Ultrasound Med Biol. 2010;36(11):1825-35.

(Reference 2) de Lédinghen V, et al. Liver Int. 2012;32(6):911-8.

(Reference 3) Tokuhara D, et al. Plos ONE. 2016;11:e0166683

(Reference 4) Isoura Y, et al. Obes Res Clin Pract. 2020;14(5):473-478

(Reference 5) Chon YE, et al. Liver Int. 2014;34(1):102-9.

• Lines 435-440: in limitations, added discussion about different CAP cutoffs

---

## [Decision Letter · Decision Letter 1]

8 Nov 2021

Screening for undiagnosed non-alcoholic fatty liver disease (NAFLD) and non-alcoholic steatohepatitis (NASH): a population-based risk factor assessment using vibration controlled transient elastography (VCTE)

PONE-D-21-19005R1

Dear Dr. Eskridge,

We’re pleased to inform you that your manuscript has been judged scientifically suitable for publication and will be formally accepted for publication once it meets all outstanding technical requirements.

Kind regards,

Daisuke Tokuhara

Academic Editor

PLOS ONE

Additional Editor Comments (optional):

I deeply appreciate authors' efforts for addressing to the reviewers' comments. Revised manuscript is well constructed. Data are well supporting the results and discussion.

Reviewers' comments:

Reviewer's Responses to Questions

**Comments to the Author**

1. If the authors have adequately addressed your comments raised in a previous round of review and you feel that this manuscript is now acceptable for publication, you may indicate that here to bypass the “Comments to the Author” section, enter your conflict of interest statement in the “Confidential to Editor” section, and submit your "Accept" recommendation.

Reviewer #2: All comments have been addressed

Reviewer #3: All comments have been addressed

2. Is the manuscript technically sound, and do the data support the conclusions?

Reviewer #2: Yes

Reviewer #3: Yes

3. Has the statistical analysis been performed appropriately and rigorously? 

Reviewer #2: Yes

Reviewer #3: Yes

4. Have the authors made all data underlying the findings in their manuscript fully available?

Reviewer #2: Yes

Reviewer #3: Yes

5. Is the manuscript presented in an intelligible fashion and written in standard English?

Reviewer #2: Yes

Reviewer #3: Yes

6. Review Comments to the Author

Reviewer #2: I deeply appreciate all of efforts of authors for revision.

The revised manuscript is well addressed to the reviewers' comments.

I have no further comments.

Reviewer #3: Revised manuscript appropriately addressed to the reviewers' comments. I have no further comments on the current version of the manuscript.

7. PLOS authors have the option to publish the peer review history of their article (what does this mean?). If published, this will include your full peer review and any attached files.

Reviewer #2: No

Reviewer #3: No

---

## [Editor Report · Acceptance letter]

18 Nov 2021

PONE-D-21-19005R1 

Screening for undiagnosed non-alcoholic fatty liver disease (NAFLD) and non-alcoholic steatohepatitis (NASH): a population-based risk factor assessment using vibration controlled transient elastography (VCTE) 

Dear Dr. Eskridge:

I'm pleased to inform you that your manuscript has been deemed suitable for publication in PLOS ONE. Congratulations! Your manuscript is now with our production department. 

Kind regards, 

on behalf of

Dr. Daisuke Tokuhara 

Academic Editor

PLOS ONE